# Deposition Behavior and Microstructure of Cold-Sprayed Ni-Coated Al Particles

Xiao Chen *, Hongkai Zhou, Zhimin Pi and Zhiwu Huang

Xinyu Key Laboratory of Materials Technology and Application for Intelligent Manufacturing, School of Mechanical and Electrical Engineering, Xinyu University, Xinyu 338004, China; zhk_1999@126.com (H.Z.); pizm_2000@126.com (Z.P.); huangzw_2022@163.com (Z.H.)
* Correspondence: chenxiaoxyxy@126.com; Tel./Fax: +86-790-6666108

**Abstract:** Cold spraying is a novel technology for preparing solid-state coatings. Single Ni-coated Al particles were deposited onto different substrates by cold spraying at different accelerating gas temperatures, as well as preparing for the coatings. The influence of the accelerating gas temperature and substrate microhardness on the particle deposition deformation, microstructure, and microhardness of Ni-coated Al coatings were investigated. The results show that the embedding depth of Ni-coated Al particles into the Al substrate increased with increasing the accelerating gas temperature. However, the cold-sprayed Ni-coated Al particles did not embed into the Q235 steel substrate, and the degree of plastic deformation of the Ni-coated Al particles increased with increasing the accelerating gas temperature. Moreover, the morphology of the Ni-coated Al splat deposited onto the Q235 steel substrate at an accelerating gas temperature of 400 °C presented a flattened morphology, which was different from the nearly spherical or ellipsoidal morphology of the Ni-coated Al feedstock. Ni-coated Al coatings exhibited the same phase compositions as the feedstock powders, and the Ni and Al phases in the coatings incurred a certain plastic deformation. Compared with the Q235 steel substrate, an Al substrate with a lower microhardness is beneficial for forming the first layer coating, as well as for the formation of an intermixing structure between the Ni-coated Al coating and Al substrate. The porosity of Ni-coated Al coatings decreased and the thickness increased when increasing the gas temperature; in particular, the coating deposited onto Al substrate had the lowest porosity and the largest thickness at an accelerating gas temperature of 400 °C. Meanwhile, the microhardness of the coating deposited onto the Al substrate was higher than that deposited onto the Q235 steel substrate under the same cold spraying conditions.

**Keywords:** Ni-coated Al; cold spray; deposition behavior; coating; microstructure

## 1. Introduction

NiAl intermetallic compounds, due to their advantageous properties, are widely applied in engineering fields [1–3]. Therefore, many researchers have prepared NiAl and its composite coatings using different surface technologies to improve the properties. NiAl and its composite coatings (e.g., $Ni_3Al$ [4], NiAl-TiC-TiB [5], $NiAl-TiO_2/Bi_2O_3$ [6], and $NiAl-ZrB_2$ [7]) have been prepared by plasma spraying technology to improve the corrosion resistance and abrasive wear properties. The addition of Fe to the NiAl-WC coating, prepared by plasma cladding technology, could reduce the amount of coating cracks. Meanwhile, remelting the refined WC particle size could also reduce the friction coefficient and wear rate [8]. The microhardness increased with the formation of NiO in the $Ni_3Al$ coatings prepared by air plasma spraying, and the wear rate and coefficient of friction decreased with the increasing NiO content in the $Ni_3Al$ coatings [9]. NiAl [10], Ni/Ni-Al [11], and $NiAl-Al_2O_3$ [12] coatings were prepared by laser cladding technology in order to improve the wear resistance property. Based on the excellent characteristics, a high velocity oxygen-fuel (HVOF) spraying process was applied to prepare the NiAl

or NiAl-composite coatings [13,14], such as nano-structured NiAl [14], β-NiAl [15], WC-Co/NiAl [16], $Al_2O_3$-NiAl [17], and $Al_2O_3$-13%$TiO_2$/NiAl [18]. Although NiAl and its composite coatings could be prepared by the above surface technologies, it is difficult to control the composition of the coatings and inhibit the oxidation phenomenon in the preparation process, which eventually affected the properties of the coating.

Cold spraying technology is a kind of novel surface technology for depositing solid-state particles at a low temperature [19,20]. Therefore, many researchers have focussed on the experimental investigation of cold sprayed metal [19–25], alloy [26,27], cermet [28–30], ceramics [31,32], and bioactive coatings [33–35]. In the aspect of NiAl and its composite coatings, due to the room temperature brittleness of NiAl intermetallic compounds, it is difficult to directly cold-spray NiAl powders to form NiAl coatings [36]. Therefore, in recent years, NiAl and its composite coatings have been prepared by cold spraying combined with post annealing treatment. Yang et al. [36] reported that ball-milled Ni/Al-$Al_2O_3$ mixture powders were deposited by cold spraying to form an Ni/Al-$Al_2O_3$ coating, and the NiAl phase was formed after post-spray annealing treatment. The key influencing factors of the dense microstructure of the NiAl-$Al_2O_3$ coating were also analyzed. Zhang et al. [37] reported that ball-milled Ni/Al mixture powders were deposited by cold spraying to form a dense Ni/Al coating, and the effect of different annealing temperatures on the phase transformation of NiAl intermetallics was investigated. Bacciochini et al. [38] reported that mechanically mixed Ni/Al powders were cold-sprayed on an aluminum 6061 substrate. The effect of different stagnation gas temperatures, substrate temperatures, and particle velocities on the compositions of the Ni/Al coatings was investigated. Jan et al. [39] studied the effect of different annealing temperatures on the density and phase transformation of a cold-sprayed Ni/Al coating. In summary, in previous studies, the Ni/Al feedstock powder for cold spraying was basically prepared using ball-milled or mechanically mixed methods. There are also few reports on the deposition behavior of Ni/Al composite particles. Meanwhile, Grujicic et al. [40] reported that, due to the influence of different powder parameters, the impact velocity of various powders would be different, so as to affect the cooperative deposition and deformation of the particles. Hence, it is difficult to obtain cooperative deposition and deformation between Ni and Al particles when cold spraying mechanically mixed Ni/Al mixture powders.

Ni-coated Al particles are a kind of powder with the structure of the Al particle coated with Ni particles. This cladding structure is beneficial in order to obtain the cooperative deposition and deformation of various phases during cold spraying. Moreover, the deposition behavior and microstructure of cold-sprayed Ni-coated Al particles have rarely been investigated. Hence, in order to further analyze the deposition behavior of cold-sprayed Ni-coated Al particles, the microstructure of the deposited Ni-coated Al particle and cold-sprayed coatings at different cold spraying parameters has been investigated in this study. The purpose of this study is to finally obtain the deformation mechanism of cold-sprayed Ni-coated Al particles and the formation mechanism of the coating. Therefore, the originality and purpose of this study are as follows: (1) Ni-coated Al powders are used as raw feedstock for cold spraying to obtain cooperative deposition and deformation; (2) the deposition behavior of cold-sprayed single Ni-coated Al particles deposited onto different substrates in different accelerating gas temperatures is investigated; and (3) the microstructure, adhesion, and microhardness of Ni-coated Al coatings deposited onto different substrates at different accelerating gas temperatures are also investigated.

## 2. Materials and Methods

### 2.1. Powder and Coating Preparation

Figure 1 shows the surface morphologies of the Ni-coated Al powders. The commercially available Ni-coated Al particles prepared by an electrolysis process (99.9 wt %; BGRIMM Technology Group Co., Ltd., Beijing, China) were used as the raw materials. The Ni coating thickness on the Al particles was about 10 μm. A laser diffraction meter (GSL-1020, Liaoning Instrument Research Institute Co., Ltd., Shenyang, China) measured

the particle size distribution of the Ni-coated Al powders (as shown in Figure 2). The size distribution (D50) of the powders was 84.4 μm. A home-made cold spray system (CS-2000, Xi'an Jiaotong University, Xi'an, China) was applied to deposit Ni-coated Al particle and coatings. The parameter details of the cold-sprayed particle and coating are listed in Table 1. Q235 steel and Al substrates polished with 0.25 μm diamond suspensions were applied to deposit the Ni-coated Al particle. Q235 steel and Al substrates blasted with 20 mesh alumina were used to cold-spray the Ni-coated Al coatings. All substrates were cut by wire electrical discharge machining (WEDM; DK7732ZG, Jiangsu Dongqing CNC Machine Tool Co., Ltd., Taizhou, China) with dimensions of 50 mm × 25 mm × 3 mm.

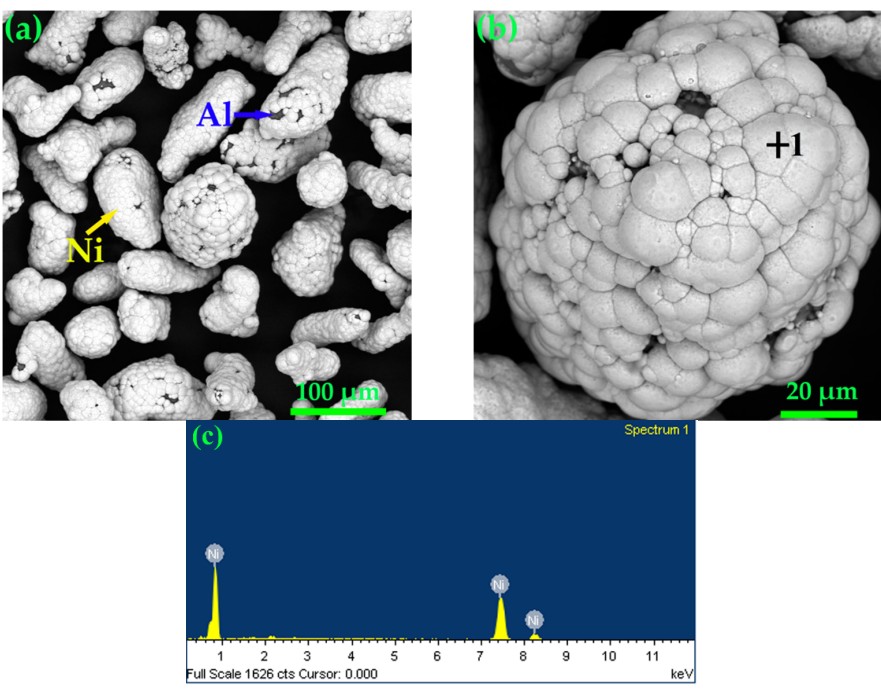

**Figure 1.** The surface morphologies of Ni-coated Al powder at (**a**) 500× and (**b**) 2000×, and (**c**) EDS analysis of the Ni coating on the Al powder.

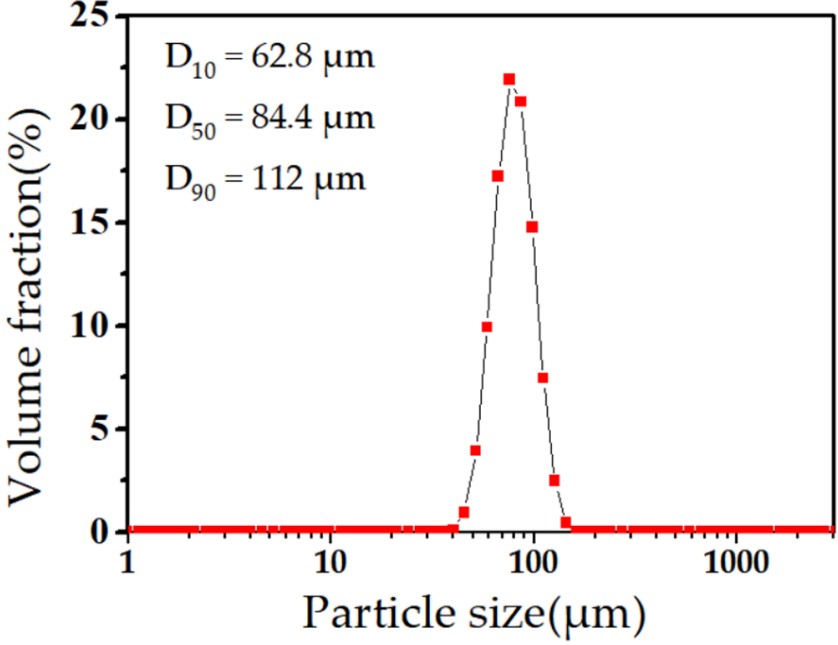

**Figure 2.** The particle size distribution of Ni-coated Al powders.

**Table 1.** The parameter details of cold spraying Ni-coated Al particle and coatings.

| Parameter | Particle | Coatings |
|---|---|---|
| Accelerating gas pressure/MPa | 2.0 | 2.0 |
| Powder-feeding gas pressure/MPa | 2.2 | 2.2 |
| Gas temperature in gun chamber/°C | $200 \pm 10, 400 \pm 10$ | $200 \pm 30, 400 \pm 30$ |
| Spray distance/mm | 20 | 20 |
| Transverse speed of gun/mm·s$^{-1}$ | 400 | 100 |
| Substrate | Q235 steel Al | Q235 steel |

### 2.2. Particle Velocity

The Ni-coated Al particle velocity distribution was measured by DPV eVOLUTION system (Tecnar Automation Ltd., Saint-Bruno-de-Montarville, QC, Canada).

### 2.3. Adhesion Strength and Microhardness

A standard tensile tester (WDW-E100, Jinan Chenda Test Machine Manufacturing Co., Ltd., Jinan, China) was used for measuring the adhesion strength of the Ni-coated Al coatings. The average of three testing values was taken as the final adhesion strength value. A digital microhardness tester (HXD-1000 TM/LCD, Shanghai Precision Instruments Co., Ltd., Shanghai, China) measured the microhardness of the Ni-coated Al coating cross-sections and substrate surfaces with a load of 2.94 N and a dwell time of 20 s. The microhardness value of all specimens was in the mean of 10 measurements. The microhardness of the Q235 steel and Al substrates was $183.6 \pm 2.3$ HV$_{0.3}$, and $32.7 \pm 1.6$ HV$_{0.3}$, respectively.

### 2.4. Microstructure Characterization

The morphologies of the feedstock particle surface, deposited particle surface, and coating cross-sections were analyzed using a scanning electron microscope (SEM; VEGA II-LSU, TESCAN, Brno, Czech Republic), respectively. The phase compositions of the feedstock powders and as-sprayed coatings were characterized by X-ray diffraction (XRD; Bruker D8 Advance, Karlsruhe, Germany) using Cu−Kα radiation (λ = 1.5418 Å, 35 kV and 35 mA). The scanning range (2θ) was 20° to 90°, and the scanning speed was 10°/min. Figure 3 shows the XRD patterns of the Ni-coated Al powders. Image analysis method (Software Image J, version 1, National Institutes of Health, Bethesda, MD, USA) was applied to calculate the content of the Ni-coated Al splats that were adhered on the substrates, the average porosity and thickness of the as-sprayed coatings, and the Ni content in the coatings.

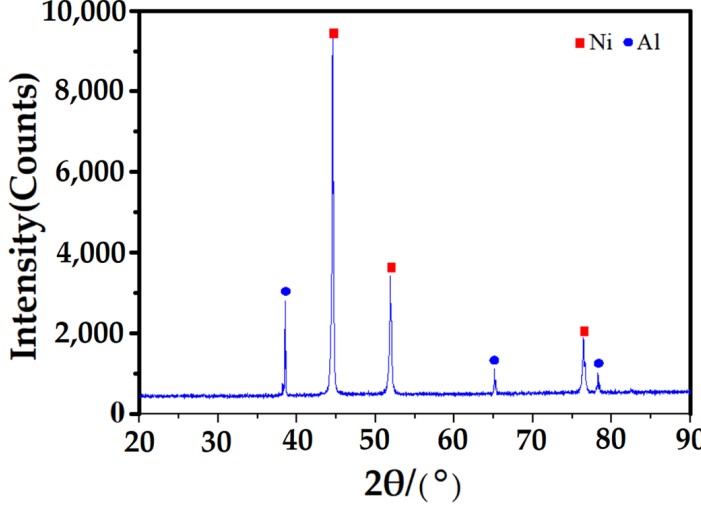

**Figure 3.** The X-ray diffraction (XRD) patterns of the Ni-coated Al powders.

## 3. Results and Discussion

### 3.1. Microstructure of Ni-Coated Al Splats

Figures 4 and 5 show the surface morphologies of the Ni-coated Al splats deposited onto Al and Q235 steel substrates at different accelerating gas temperatures, respectively. It can be seen that the Ni-coated Al splat was partially embedded into the Al substrate at an accelerating gas temperature of 200 °C (as shown in Figure 4a), and the Ni-coated Al splat exhibited the same morphology with the original feedstock powder (as shown in Figure 1). Moreover, gaps (as indicated by purple arrow in Figure 4a) appeared between the splat and substrate, and due to the impact deformation between the splat and substrate, revers phenomenon (as indicated by the cyan arrow in Figure 4a) appeared on the Al substrate around the splat. When increasing the accelerating gas temperature to 400 °C, it was found that the Ni-coated Al splat was basically completely embedded into the Al substrate (as shown in Figure 4b). Meanwhile, the plastic deformation of the Al substrate was more severe, and in addition to the gap formation (as indicated by the purple arrow in Figure 4b); the revers phenomenon (as indicated by cyan arrow in Figure 4b) was also serious. Moreover, due to the effect of the impact of the particle on the Al substrate, there appeared obvious ripple (as indicated by green arrow in Figure 4b) on Al substrate around the splat. However, with the increasing of substrate microhardness (Q235 steel), Ni-coated Al particle exhibited the same morphology with the feedstock powder did not embed into Q235 steel substrate in the accelerating gas temperature of 200 °C (as shown in Figure 5a), and due to high-velocity effect of the impact, the peeling off (as indicated by the black arrow in Figure 5a) and falling off (as indicated by the blue arrow and blue dotted rink in Figure 5a) phenomena of the Ni phase appeared on the surface of the Ni-coated Al splat. However, as the accelerating gas temperature increased to 400 °C, severe plastic deformation of Ni-coated Al particle occurred (as shown in Figure 5b). The morphology of the Ni-coated Al splat changed from an ellipsoidal or nearly spherical shape to a flattened shape, and warping phenomenon (as indicated by orange arrow in Figure 5b) appeared at the edge of the splat. Li et al. [19] reported that particle deposition deformation was affected by different gas temperature and substrate microhardness. Therefore, due to the difference of microhardness between the Al and Q235 steel substrates, different morphologies of the Ni-coated Al were found for the different accelerating gas temperatures after their impact on the substrates in this study. Yin et al. [41] also reported that particle velocity and particle deformation would be increased by increasing the gas temperature. Therefore, the velocity of the cold-sprayed Ni-coated Al particles was measured using a DPV eVOLUTION system in this study. Figure 6 shows the velocity distribution of the Ni-coated Al particles. It could be found that the average velocity of the Ni-coated Al particles at an accelerating gas temperature of 200 °C and 400 °C was 543 m/s and 601 m/s, respectively. This illustrates that the velocity of the Ni-coated Al particles increased as the accelerating gas temperature increased. Meanwhile, the penetration depth of the splat deposited on the Al substrate increased as the velocity increased. With the increasing substrate microhardness (Q235 steel), the particle deformation increased with the increasing particle velocity.

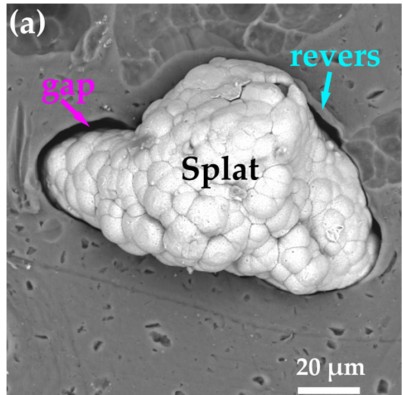 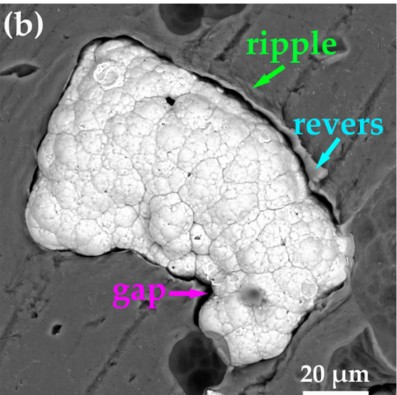

**Figure 4.** Surface morphologies of Ni-coated Al splat deposited onto the Al substrate at an accelerating gas temperature of (**a**) 200 °C and (**b**) 400 °C.

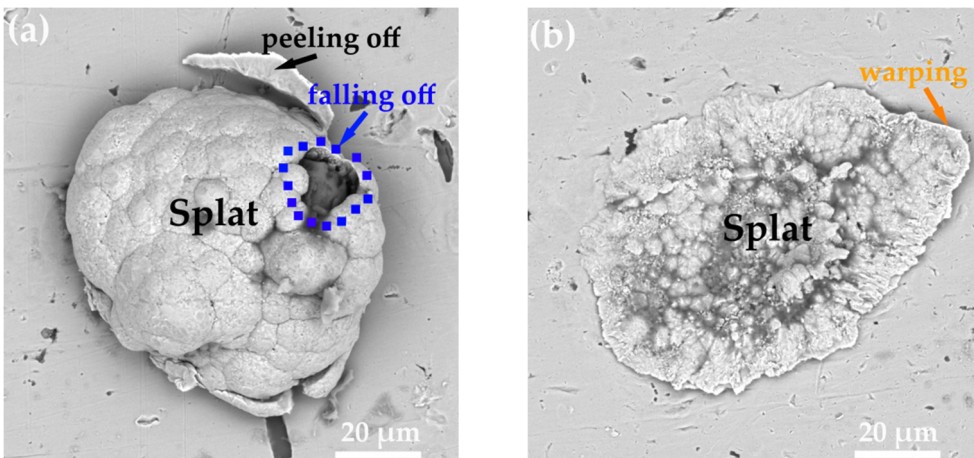

**Figure 5.** Surface morphologies of Ni-coated Al splat deposited onto a Q235 steel substrate at an accelerating gas temperature of (**a**) 200 °C and (**b**) 400 °C.

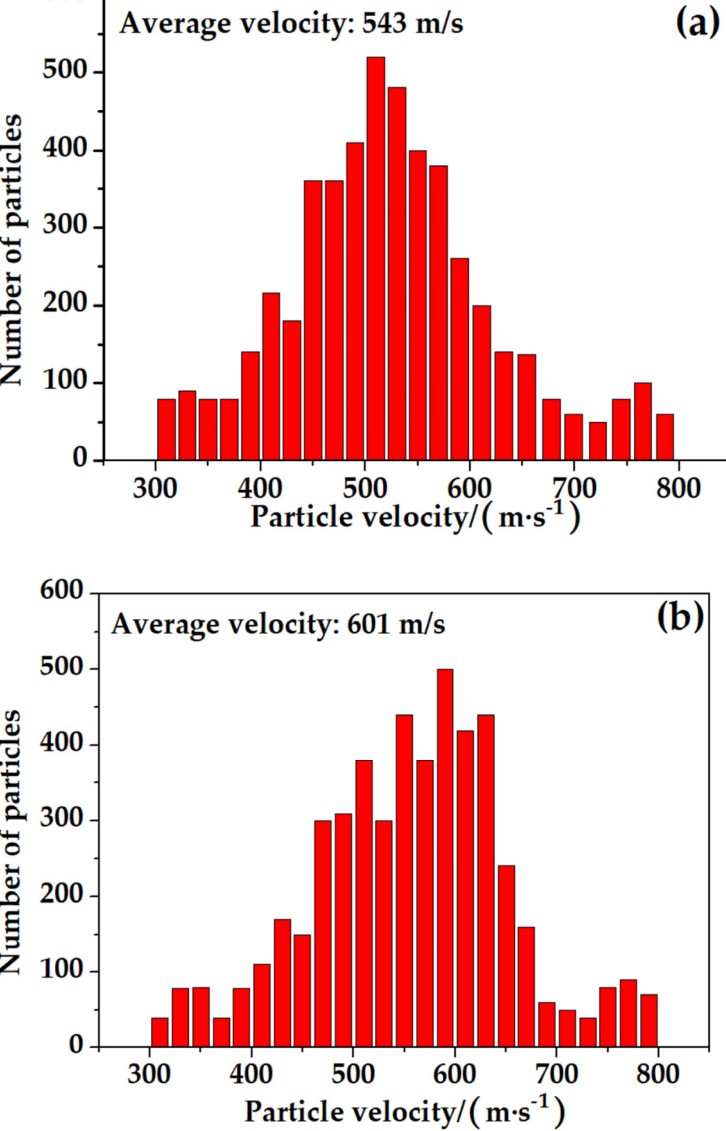

**Figure 6.** Velocity distribution of Ni-coated Al particles at an accelerating gas temperature of (**a**) 200 °C and (**b**) 400 °C.

### 3.2. Microstructure of Ni-Coated Al Coatings

Figure 7 shows the cross-section morphologies of the Ni-coated Al coatings deposited onto the Al substrate at an accelerating gas temperature of 200 °C and 400 °C. It was found that, due to the subsequent particles lacking an effect of impact on the surface layer particles, the surface layer Ni-coated Al particles basically retained the original morphology in the coating (as indicated by the green dotted rink in Figure 7). Meanwhile, there was no obvious interface between the Ni-coated Al coating and Al substrate, and an intermixing phenomenon appeared between the coating and substrate (as indicated by the cyan arrow in Figure 7); in particular, the accelerating gas temperature at 200 °C displayed more of an intermixing phenomenon. The main reason for the intermixing phenomenon is that, due to the low deposition efficiency, as the incident particle was deposited on the surface of the soft substrate, the fracture of the hard phase was induced by the rebounding particle constantly impacting the first-layer coating, and the fracture fine phase was embed into the soft substrate to form an intermixing interface [42]. Therefore, by analyzing the bonding morphologies of the Ni-coated Al particles deposited on the Al substrate surface at different accelerating gas temperatures (as shown in Figure 8), it could be found that craters formed as the incident particles rebounded from the Al substrate surface (as indicated by the white arrow in Figure 8). The ratio of Ni-coated Al splats that were adhered on the Al substrate surface (as indicated by the black arrow in Figure 8) was 3.17% at an accelerating gas temperature of 200 °C. However, the ratio of splats that were adhered on the Al substrate surface increased to 38.24% after increasing the accelerating gas temperature to 400 °C. This illustrates that increasing the accelerating gas temperature was beneficial to the deposition of particles on the substrate surface. Therefore, in addition to more sever plastic deformation (as indicated by the purple square frame and yellow arrow in Figure 7a), more rebounding particles impacted the first layer coating to form a greater intermixing interface at an accelerating gas temperature of 200 °C. However, due to the increasing deposition efficiency, less of the intermixing phenomenon appearing between the coating and Al substrate as the accelerating gas temperature increased to 400 °C. Meanwhile, due to the effect of the impact on the subsequent incident particles, streamlined morphologies were distributed in the coating (as indicated by the yellow arrow in Figure 7). By measuring the porosity and thickness of the Ni-coated Al coatings deposited onto the Al substrate, it was found that the porosity of the coatings sprayed at an accelerating gas temperature of 200 °C and 400 °C was 1.41% and 0.33%, respectively. The thickness of the coatings was 64.7 and 140.6 μm, respectively.

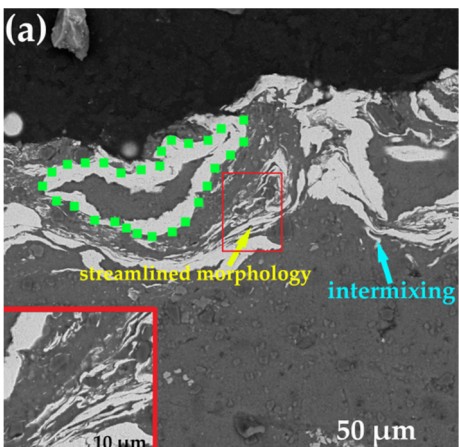 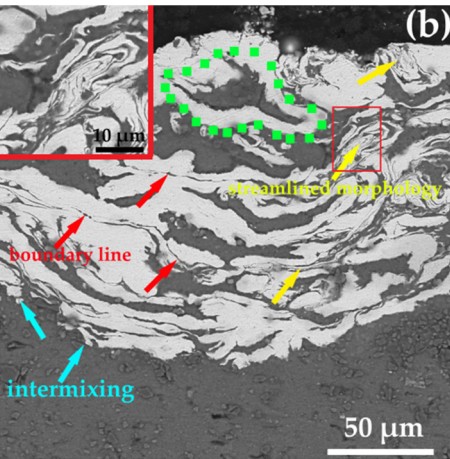

**Figure 7.** Cross-section morphologies of Ni-coated Al coatings deposited onto an Al substrate at an accelerating gas temperature of (**a**) 200 °C and (**b**) 400 °C.

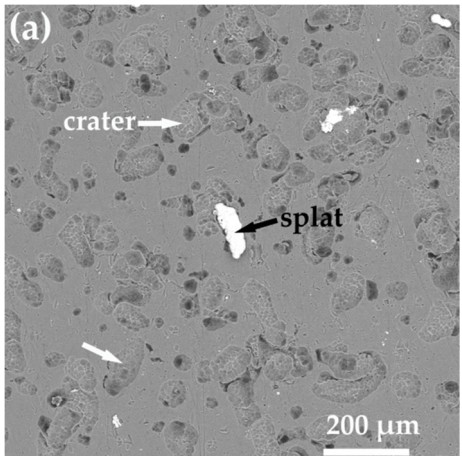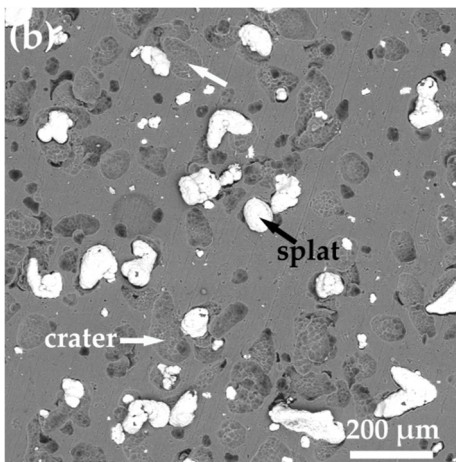

**Figure 8.** Morphologies of Ni-coated Al particles deposited onto an Al substrate at an accelerating gas temperature of (**a**) 200 °C and (**b**) 400 °C.

Figure 9 shows the cross-section morphologies of the Ni-coated Al coatings deposited onto the Q235 steel substrate at an accelerating gas temperature of 200 °C and 400 °C. An obvious interface with no micro-crack was observed between the coating and substrate (as indicated by the purple arrow in Figure 9). Meanwhile, there was no impact from the Ni-coated Al particles embedding into the Q235 steel substrate. Combined with the analysis results of the deposition deformation of Ni-coated Al particles (as shown in Figure 5), the morphology of the deposited particles in the coating was close to the original particles at an accelerating gas temperature of 200 °C (as shown in Figure 9a). However, the severe plastic deformation impacting the Ni-coated Al particles in the coating occurred at an accelerating gas temperature of 400 °C. Meanwhile, the morphology impacting the Ni-coated Al particles on the coating changed from an ellipsoidal or nearly spherical shape to a flattened shape (as shown in Figure 9b). This phenomenon is consistent with the analysis results of the deposition behavior of Ni-coated Al particles deposited onto the Q235 steel substrate (as shown in Figure 5b). By analyzing the bonding morphologies of the Ni-coated Al particles deposited onto the Q235 steel substrate surface at different accelerating gas temperatures (as shown in Figure 10), it was found that there were no obvious craters observed from the Q235 steel substrate. Compared with the ratio of splats adhered on the Q235 steel substrate surface at an accelerating gas temperature of 200 °C, the ratio of splats was more at an accelerating gas temperature of 400 °C (as shown in Figure 10b), but less than that of the splats adhered on the Al substrate surface under the same spraying conditions (as shown in Figure 8b). Gao et al. [43] reported that the absorbing degree of the energy of the high velocity particles was influenced by substrates with a different microhardness, and led to the ratio of splats that were adhered on the substrate being different. Therefore, compared with the Al substrate, the Q235 steel substrate could not absorb more energy from high velocity particles, and thus led to a lower ratio of splats that were adhered on the Q235 steel substrate surface. By further analyzing the cross-section of the morphologies of the Ni-coated Al coatings, it was found that a boundary line was observed between the Ni phases of the adjacent deposited Ni-coated Al particles (as indicated by red arrow in Figures 7 and 9), and there were few metallurgical bonding phenomena between the Ni-coated Al splats [19]. The major cause for this is that, due to the plastic flow of the Al phase accompanied with the movement of the Ni phase during the impact of particles, this could lead to reducing the heat quantity during the impact between Ni-coated Al particles. Meanwhile, due to the effect of the impact of the rebounding of the Ni-coated Al particles, plastic deformation occurred on the surface of the Q235 steel substrate during the formation of the first-layer coating (as indicated by the orange arrow in Figure 9). By measuring the porosity and thickness of the Ni-coated Al coatings deposited onto the Q235 steel substrate, it was found that the porosity of the coatings sprayed at an accelerating gas temperature of 200 °C and 400 °C was 1.63% and 0.68%, respectively. The thickness

of the coatings was 102.4 μm and 110.9 μm, respectively. Compared with the porosity and thickness of the Ni-coated Al coatings deposited onto the Al substrate, it was found that the porosity of the coatings deposited onto the Q235 steel substrate increased, and the thickness of the coating sprayed at an accelerating gas temperature of 200 °C was larger than that of the coating deposited onto the Al substrate under the same spraying conditions. Yin et al. [44] reported that substrate hardness has a certain influence on the thickness of the coating, the soft substrate is beneficial to form the first layer coating, which led to obtain larger thickness of the coating. Therefore, an Al substrate with a low microhardness was beneficial to form the first layer of the Ni-coated Al coating under the same spraying conditions. Meanwhile, the deposited coating was constantly impacted by the subsequent incident particles, so that the porosity of the Ni-coated Al coatings deposited on the Al substrate was lower than that of the coating deposited onto the Q235 steel substrate. Due to more particles rebounding from the harder substrate, the first layer coating deposited onto the Q235 steel substrate was relatively difficult, so that the thickness of the coating was smaller than that deposited onto the Al substrate at an accelerating gas temperature of 400 °C. However, at an accelerating gas temperature at 200 °C, due to the lower deposition efficiency and the lack of an impact effect on the subsequent incident particles, the last deposited particles close to the original particle morphology (as shown in Figure 9a) were only mechanically bonded to the surface of the Q235 steel substrate, which led to the thickness of the coating deposited onto the Q235 steel substrate being larger than that of the coating deposited onto the Al substrate at an accelerating gas temperature of 200 °C.

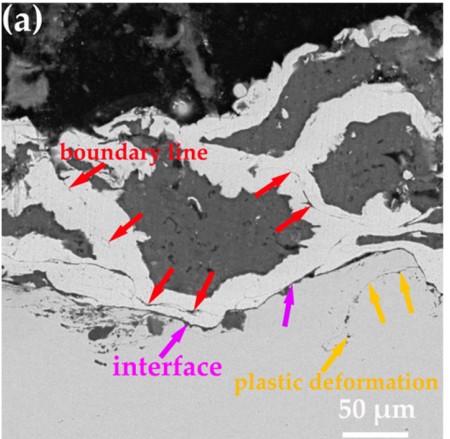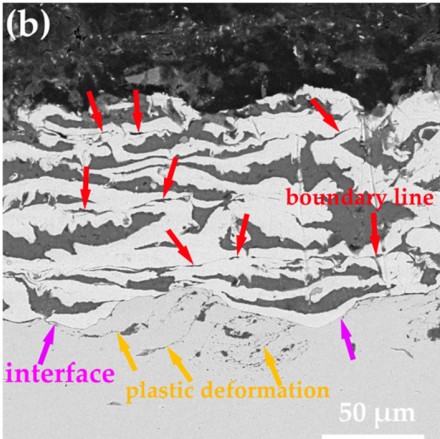

**Figure 9.** Cross-section morphologies of Ni-coated Al coatings deposited onto a Q235 steel substrate at an accelerating gas temperature of (**a**) 200 °C and (**b**) 400 °C.

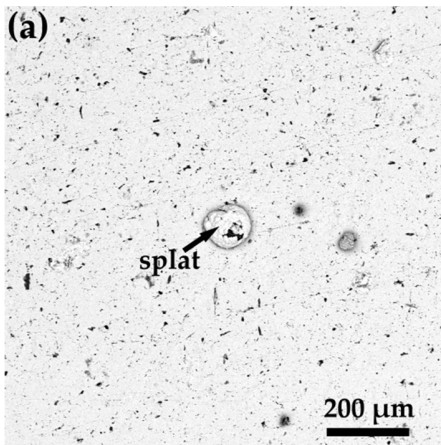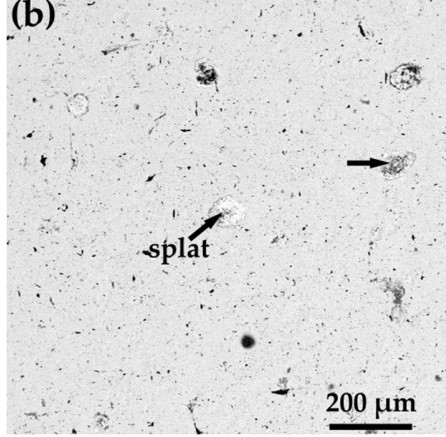

**Figure 10.** Morphologies of Ni-coated Al particles deposited onto a Q235 steel substrate at an accelerating gas temperature of (**a**) 200 °C and (**b**) 400 °C.

### 3.3. XRD Patterns of Ni-Coated Al Coatings

Figure 11 shows the XRD patterns of the Ni-coated Al coatings deposited onto the Al and Q235 steel substrates at an accelerating gas temperature of 200 °C and 400 °C. Compared with the XRD patterns of the original powders, it can be seen that the phase compositions of the as-sprayed coatings were consistent with the original particles. Due to low-temperature deposition characteristics of cold spraying, no chemical reaction occurred between the phases of Ni-coated Al particles, so that the phase compositions of the powders were transplanted into the coatings [21]. By measuring the full widths at half maximums (FWHMs) of the Ni and Al diffraction peaks of the powders and coatings (as shown in Figure 12), it was found that the FWHMs of Ni and Al diffraction peaks of the Ni-coated Al powders were smaller than those of the coatings. Moreover, the FWHMs of the Ni and Al diffraction peaks of the Ni-coated Al coatings as-sprayed at an accelerating gas temperature of 400 °C were larger than those of the coatings as-sprayed at an accelerating gas temperature of 200 °C. This further illustrates that a certain plastic deformation occurred on the Ni-coated Al particles during cold spraying. Moreover, the plastic deformation of particles was more serious with the increase in accelerating gas temperature. Furthermore, the FWHMs of Ni and Al diffraction peaks of the coatings deposited onto the Q235 steel substrate were smaller than those of the coatings deposited onto the Al substrate. The FWHMs analysis results were consistent with the microstructure analysis of the coatings.

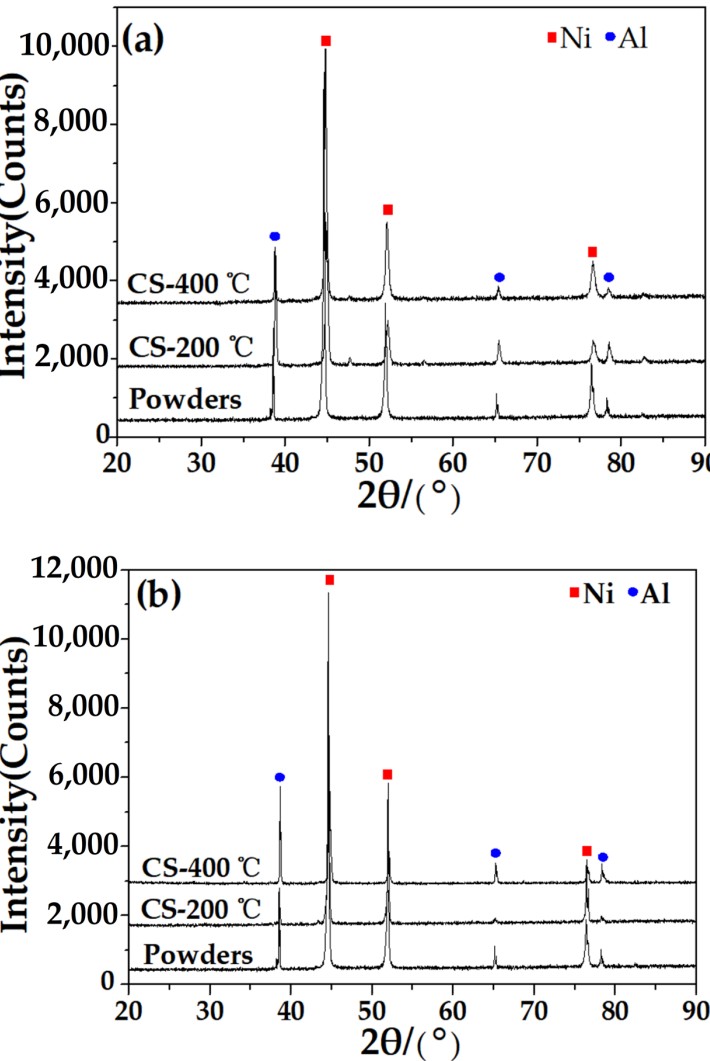

**Figure 11.** XRD patterns of Ni-coated Al coatings deposited onto an (**a**) Al substrate and (**b**) Q235 steel substrate.

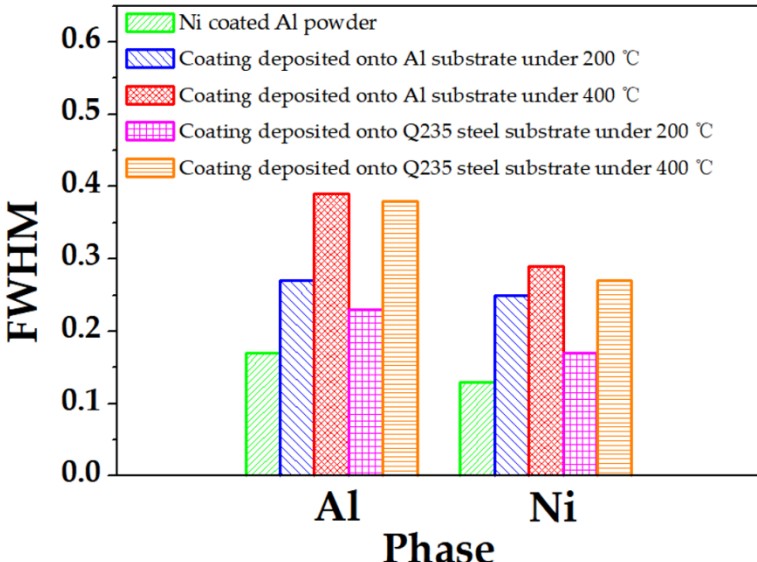

**Figure 12.** Full widths at half maximum (FWHMs) of Ni and Al in Ni-coated Al powders and coatings.

*3.4. Adhesion Strength and Microhardness of Ni-Coated Al Coatings*

Figure 13 shows the adhesion strength of the Ni-coated Al coatings deposited onto the Al and Q235 steel substrates at different accelerating gas temperatures. The mean adhesion strength of the Ni-coated Al coatings deposited onto the Al and Q235 steel substrates at an accelerating gas temperature of 200 °C and 400 °C was 41.6 ± 3.1 and 34.8 ± 1.6 MPa, and 21.3 ± 1.5 and 31.5 ± 1.3 MPa, respectively. It was found that the adhesion strength of the coatings deposited onto the Al substrate was larger than that of the coatings deposited onto the Q235 steel substrate. This also revealed that the intermixing interface between the Ni-coated Al coating and Al substrate was beneficial for increasing the adhesion strength. Compared with the adhesion strength of the Ni-Al coatings prepared by plasma spraying [45], the adhesion strength of the Ni-coated Al coatings deposited on the Al substrate was larger in this study. This is further illustrates that the well-bonded interface between the coating and substrate could increase the adhesion strength. Meanwhile, due to the increased intermixing interface phenomenon, the adhesion strength of the coating deposited Al substrate at an accelerating gas temperature of 200 °C was also larger than that of the coatings deposited onto the Al substrate at an accelerating gas temperature of 400 °C.

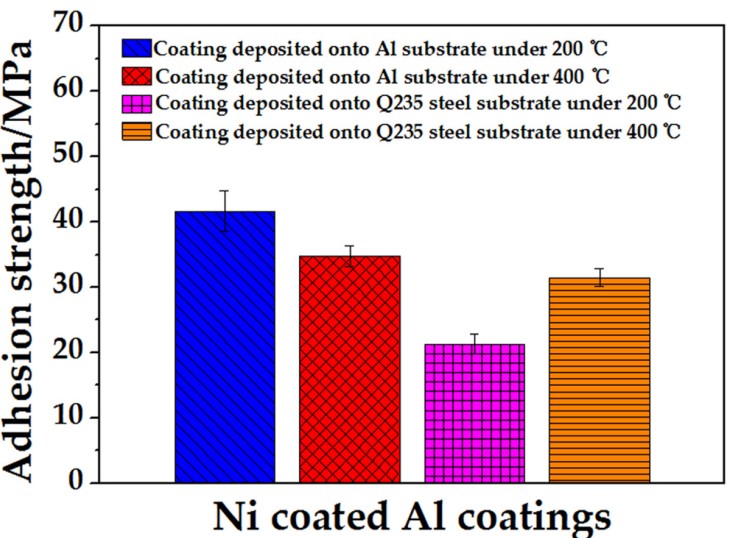

**Figure 13.** Adhesion strength of Ni-coated Al coatings deposited onto different substrates at different accelerating gas temperatures.

Figure 14 shows the average microhardness of the Ni-coated Al coatings deposited onto the Al and Q235 steel substrates at different accelerating gas temperatures. The mean microhardness of the Ni-coated Al coatings deposited onto the Al and Q235 steel substrates at an accelerating gas temperature of 200 °C and 400 °C was $151.6 \pm 8.1$ and $228.2 \pm 10.5$ HV$_{0.3}$, and $136.7 \pm 7.8$ and $191.6 \pm 9.3$ HV$_{0.3}$, respectively. It was found that the microhardness of the as-sprayed coatings increased when increasing the accelerating gas temperature. This is mainly attributed to the higher Ni content in the coating at an accelerating gas temperature of 400 °C. By measuring the Ni content in the coatings (as shown in Figure 15), it was also seen that the Ni content in the coatings deposited onto the Al substrate was higher than that in the coatings deposited onto the Q235 steel substrate. The major cause for this phenomenon is that, according to the analysis results of the microstructure of the Ni-coated Al coatings, increasing the thickness of the Ni-coated Al coating deposited onto the same substrate at a higher accelerating gas temperature would increase the Ni content. Meanwhile, an Al substrate with a low microhardness was beneficial for forming the first layer of the Ni-coated Al coating under the same spraying conditions, and when an intermixing interface occurred between the Ni-coated Al coating and the Al substrate, the Ni content in the coating deposited onto the Al substrate was more than that in the coating deposited onto the Q235 steel substrate. Therefore, the microhardness of the coatings deposited onto the Al substrate was higher than that of the coatings deposited onto the Q235 steel substrate under the same spraying conditions. Compared with the microhardness of the Ni/Al coatings by depositing mechanically mixed Ni/Al mixture powders [46], the microhardness of the Ni-coated Al coatings was higher in this study.

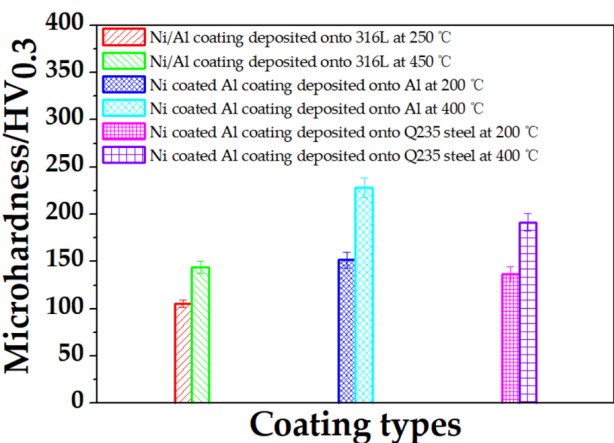

**Figure 14.** Microhardness of the coatings deposited onto different substrates in different accelerating gas temperatures.

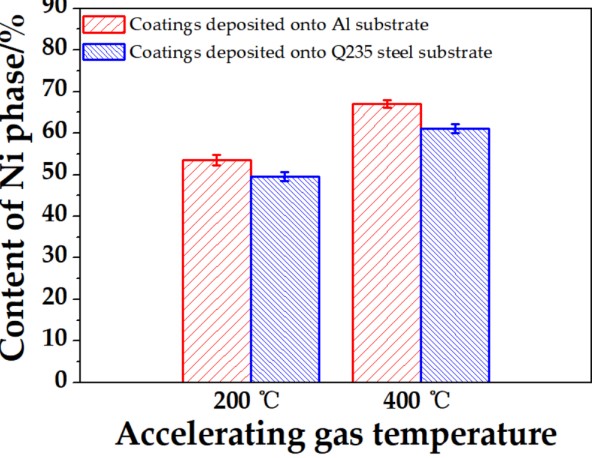

**Figure 15.** The content of the Ni phase in the coatings deposited onto different substrates at different accelerating gas temperatures.

## 4. Conclusions

Ni-coated Al powder was successfully deposited onto Al and Q235 steel substrates by a cold-spraying process at an accelerating gas temperature of 200 °C and 400 °C to form Ni-coated Al coatings. The deposition behavior of the deposited Ni-coated Al particles and the microstructure of the as-sprayed coatings were investigated. According to the results in this study, the following conclusions can be drawn:

- The size distribution (D50) of the Ni-coated Al powders with an ellipsoidal or nearly spherical morphology was 84.4 μm. The phase compositions of the Ni-coated Al powder were Ni and Al phases.
- By analyzing the deposition behavior of cold-spraying Ni-coated Al particles, Ni-coated Al splat deposited onto the Al substrate exhibited the same morphology as the original feedstock powder. By increasing the accelerating gas temperature, the embedding depth of the splat deposited onto the Al substrate increased, and the plastic deformation of the Al substrate (e.g., gap, revers, and ripple) was more severe.
- Due to the higher microhardness of the Q235 steel substrate, there was no plastic deformation that occurred on the Q235 steel substrate surface. At an accelerating gas temperature of 200 °C, the Ni-coated Al particle deposited on Q235 steel substrate exhibited an ellipsoidal or nearly spherical morphology with a peeling off and falling off phenomenon. As the accelerating gas temperature increased to 400 °C, the morphology of the Ni-coated Al splat changed from an ellipsoidal or nearly spherical shape to a flattened shape, and warping phenomenon appeared at the edge of the splat.
- Due to the lower microhardness of the Al substrate, an intermixing phenomenon appeared between the coating and Al substrate. Compared with the accelerating gas temperature of 400 °C, there was an increased intermixing interface observed between the coating and Al substrate at an accelerating gas temperature of 200 °C owing to the lower deposition efficiency. The soft Al substrate was beneficial to form the first layer coating, and the ratio of splats adhered on the Al substrate surface at an accelerating gas temperature of 200 °C and 400 °C was 3.17% and 38.24%, respectively. The porosity of the coatings deposited onto the Al substrate at an accelerating gas temperature of 200 °C and 400 °C was 1.41% and 0.33%, respectively, and the thickness of the coatings was 64.7 μm and 140.6 μm, respectively.
- Due to the higher microhardness of the Q235 steel substrate, there was no impact of Ni-coated Al particles embedded into the substrate. Meanwhile, no intermixing phenomenon appeared between the coating and Q235 steel substrate, and it was difficult to form the first layer coating. Due to more rebounding particles, plastic deformation occurred on the Q235 steel substrate surface. The morphology of the deposited particles was closed to the original particles at an accelerating gas temperature of 200 °C. The plastic deformation of the deposited particles was more severe at an accelerating gas temperature of 400 °C, and the morphology of the impacted particles changed from an ellipsoidal or nearly spherical shape to a flattened shape. The porosity of the coatings deposited onto the Q235 steel substrate at an accelerating gas temperature of 200 °C and 400 °C was 1.63% and 0.68%, respectively, and the thickness of the coatings was 102.4 μm and 110.9 μm, respectively.
- The phase compositions of all of the coatings were Ni and Al phases, which were the same as for the original powders. According to FWHMs results of the Ni and Al diffraction peaks of Ni-coated Al powders and coatings, a certain plastic deformation occurred on the Ni-coated Al particles during cold spraying. The plastic deformation of the Ni-coated Al particles was more serious with an increase in accelerating gas temperature, especially the particles deposited onto the Al substrate.
- The mean adhesion strength of the Ni-coated Al coatings deposited onto the Al and Q235 steel substrates at an accelerating gas temperature of 200 °C and 400 °C was 41.6 ± 3.1 and 34.8 ± 1.6 MPa and 21.3 ± 1.5 and 31.5 ± 1.3 MPa, respectively.
- The microhardness of the Ni-coated Al coatings increased with the increase in the accelerating gas temperature. The mean microhardness of the Ni-coated Al coatings

deposited onto the Al and Q235 steel substrates at an accelerating gas temperature of 200 °C and 400 °C was 151.6 ± 8.1 and 228.2 ± 10.5 $HV_{0.3}$ and 136.7 ± 7.8 and 191.6 ± 9.3 $HV_{0.3}$, respectively. Due to the higher Ni content in the coatings deposited onto the Al substrate, the microhardness of the coatings deposited onto the Al substrate was higher than that of the coatings deposited onto the Q235 steel substrate.

**Author Contributions:** Conceptualization, X.C.; methodology, X.C. and H.Z.; software, H.Z.; validation, X.C. and Z.P.; formal analysis, X.C.; investigation, Z.H.; resources, X.C.; data curation, X.C. and Z.P.; writing—original draft preparation, X.C.; writing—review and editing, Z.H.; visualization, Z.P.; supervision, H.Z.; project administration, X.C.; funding acquisition, X.C. All authors have read and agreed to the published version of the manuscript.

**Funding:** This research was funded by the National Science Foundation of China, grant number 52161018; the Science and Technology Project of Jiangxi Educational Bureau, grant number GJJ212304; and the Science Technology Project of Jiujiang City, grant number [2015] 64.

**Institutional Review Board Statement:** Not applicable.

**Informed Consent Statement:** Not applicable.

**Data Availability Statement:** Not applicable.

**Conflicts of Interest:** The authors declare no conflict of interest.

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
