# Peer review of "Deposition Behavior and Microstructure of Cold-Sprayed Ni-Coated Al Particles"

_coatings, doi:10.3390/coatings12040544_

Round 1
Reviewer 1 Report
1. Have the authors evaluated the coating-substrate adhesion in the mentioned cases? In any case, some comments are needed considering the shown results in the paper.
Thanks very much for your thoughtful suggestion. Your suggestion for us is very valuable. We added the experiment about adhesion strength of Ni coated Al coatings in this study. Relevant contents are also added to the revision.
A standard tensile tester (WDW-E100, Jinan Chenda Test Machine Manufracturing Co., Ltd., Shangdong, China) was used for measuring the adhesion strength of Ni coated Al coatings. The average of three testing values is taken as the final adhesion strength value.
Figure 13 shows adhesion strength of Ni coated Al coatings deposited onto Al and Q235 steel substrates in different accelerating gas temperature. The mean adhesion strength of Ni coated Al coatings deposited onto Al and Q235 steel substrates in accelerating gas temperature of 200 ℃ and 400 ℃ was 41.6 ± 3.1 and 34.8 ± 1.6 MPa, 21.3± 1.5 and 31.5 ± 1.3 MPa, respectively. It can be found that adhesion strength of the coatings deposited onto Al substrate was larger than that of the coatings deposited onto Q235 steel substrate. This also revealed that intermixing interface between Ni coated Al coating and Al substrate was beneficial to increase adhesion strength. Compared with adhesion strength of Ni-Al coatings prepared by plasma spraying, the adhesion strength of Ni coated Al coatings deposited on Al substrate was larger in this study. This is further illustrated that well-bonded interface between the coating and susbtrate could increase adhesion strength. Meanwhile, due to more intermixing interface phenomenon, adhesion strength of the coating deposited Al substrate at the accelerating gas temperature of 200 ℃ was also larger than that of the coatings deposited onto Al substrate at the accelerating gas temperature of 400 ℃.
2. Please explain why the Ni-content in the coating is higher when an Al substrate is used instead of a steel one.
Thanks very much for your thoughtful suggestion. Relevant contents are also added to the revision.
The major cause for this phenomenon is that, according to the analysis results of microstructure of Ni coated Al coatings, increasing the thickness of Ni coated Al coating deposited onto the same substrate at higher accelerating gas temperature would increase Ni content. Meanwhile, Al substrate with low microhardness was beneficial to form the first layer Ni coated Al coating under the same spraying conditions, and intermixing interface occurred between Ni coated Al coating and Al substrate, it also led to obtain Ni content in the coating deposited onto Al substrate was more than that in the coating deposited onto Q235 steel substrate.
Special thanks to you for your good comments. We hope that these revisions are satisfactory and that the revised version will be acceptable for publication in Coatings.
Thank you very much for your work concerning our paper.

Reviewer 2 Report
1. The key results of the presented study are expected for cold sprayed structures. For example, it is well known that particle deformation is increased with increasing gas temperature, or, the penetration depth of the deposited particles increases with decreasing hardness of the substrate, or, particle deformation increases as increasing particle velocity, etc. Therefore, the originality and purpose of this study should be stated more clearly.
Thanks very much for your thoughtful suggestion. You are right. Your suggestion for us is very valuable. According to your suggestion, we explain the originality and purpose of this study in the introduction. The originality and purpose of this study is as follow: (1) Ni coated Al powders were used as raw feedstock for cold spraying to obtain cooperative deposition and deformation; (2) the deposition behavior of cold-sprayed single Ni coated Al particles deposited onto different substrates in different accelerating gas temperature was investigated; (3) the microstructure, adhesion, and microhardness of Ni coated Al coatings deposited onto different substrates in different accelerating gas temperature were also investigated.
2. Some paper published in 2021-2022 (such as doi.org/10.1016/j.pmatsci.2021.100839 or doi.org/10.1134/S2070205118020168 etc.) can be considered in the introduction section in the second paragraph that gives information about cold-spray technology and its application. Also, some discussion can be detailed.
Thanks very much for your thoughtful suggestion. We added the published paper in the revision.
- Poza P.; Garrido-Maneiro M. Á. Cold-sprayed: Microstructure, mechanical properties, and wear behavior. Prog. Mater. Sci., 2022, 123, 100839.
- Dikici B.; Topuz M. Production of annealed cold-sprayed 316L stainless steel coatings for biomedical applications and their invitro corrosion response. Prot. Met. Phys. Chem., 2018, 54, 333-339.
3. Which technique was used in the Ni coating of the Al powders?
Thanks very much for your thoughtful question. The commercially available Ni coated Al particles prepared by electrolysis process.
4. If available, EDS mapping may be better to prove the Ni coating on the Al powders.
Thanks very much for your thoughtful suggestion. You are right. We supplemented relevant contents in the revision.
5. How was selected/determined spray distance as 20mm?
Thanks very much for your thoughtful question. We have investigated the coatings as-sprayed in different spray distance (e.g., 10mm, 20mm, and 30mm), and found that the deposition effect was better as selecting spray distance as 20 mm. Hence, we determined spray distance as 20 mm in this study.
6. Authors' argument on page 4, line 125: "... Ni content in 124 the coatings were calculated by image analysis method .." how was measured the Ni content of the particles by image analysis?
Thanks very much for your thoughtful question. Ni content measured by Software Image J was mainly calculated by measuring the proportion of the area of Ni phase in the total area.
7. The text color tones in Fig. 7 should be redrawn, the illustration of the Figs may cause problems for readers.
Thanks very much for your thoughtful question. We redrew Fig.7 in the revision.
8. More scientific writing language should be used throughout the manuscript. For example, ''…plastic deformation of Al substrate was more serious…'' what does it mean "more serious"? or "...due to the extrusion effect of the particle on the Al substrates...""extrusion effect?"
Thanks very much for your thoughtful suggestion. Your suggestion for us is very valuable. We rewrote some sentences in the revision.
Special thanks to you for your good comments. We hope that these revisions are satisfactory and that the revised version will be acceptable for publication in Coatings.
Thank you very much for your work concerning my paper.

Reviewer 3 Report
1. Mention the few application of the Cold spray deposition of Ni/Al particles on different substrate.
Thanks very much for your thoughtful suggestion. According to the literatures, there are few reports of the cold spray deposition of Ni/Al particles on different substrates.
2. Ni coated Al particles are prepared through electrolysis process or any other process, please indicate and If possible, mention the Ni coating thickness on Al particles.
Thanks very much for your thoughtful suggestion. The commercially available Ni coated Al particles prepared by electrolysis process. The Ni coating thickness on Al particles was about 10 μm.
3. On what basis selected the 200 deg.C and 400 deg. C during cold spray?
Thanks very much for your thoughtful question. In a previous study, we have studied the deposition behavior of cold-sprayed single particle (e.g., WC-Co, Ti, HA, HA/Ti, Ni, etc) in different accelerating gas temperature. Meanwhile, we also have investigated the deposition behavior of cold-sprayed Ni coated Al particle in different accelerating gas temperature. Hence, according to the previous study results and experience, two kinds of temperature (200 deg.C and 400 deg. C) was chosen based on the obviously different of deposition behavior of Ni coated Al particle.
4. Fig. 4 indicate the wettability of the particles are very poor, please justify it.
Thanks very much for your thoughtful suggestion. Due to the impacting effect of high-velocity particle on the Al substrate, the particle embedded into Al substrate, as well as the plastic deformation of Al substrate occurred, and in addition to obvious gaps formation appeared on Al substrate around the splat.
Special thanks to you for your good comments. We hope that these revisions are satisfactory and that the revised version will be acceptable for publication in Coatings.
Thank you very much for your work concerning my paper.

Round 2
Reviewer 2 Report
Thank you for the revisions on the manuscript.